# Evaluation Using Polar Coordinate of the Representation of Movement in the Drawings of Children Aged 5 to 8 Years

**DOI:** 10.3390/ijerph19052844

**Published:** 2022-03-01

**Authors:** Maria Luz Urraca-Martínez, Maria Teresa Anguera, Sylvia Sastre-Riba

**Affiliations:** 1Department of Educational Sciences, University of La Rioja, 26006 Logroño, Spain; silvia.sastre@unirioja.es; 2Faculty of Psychology, Institute of Neurosciences, University of Barcelona, 08035 Barcelona, Spain; tanguera@ub.edu

**Keywords:** evaluation, mental representation, neuroconstructivism, movement drawing, infancy, polar coordinates

## Abstract

The progressive complexity of mental representation is the basis for changes in human cognitive development. Evaluation of its external manifestations as graphic representation in drawings could be an instrument to understand changes in cognitive development and representational complexity. The aim of this study is to evaluate the appearance and role of the indicators used by children to represent moving figures in their drawings. This allows us to know the continuum from its non-manifestation to full expression through the vectorial interrelationships of the graphic indicators in each of the ages studied. Participants were n = 240 children from 5 to 8 years old; their drawings of two moving figures were analyzed, applying the polar coordinate technique. Results show a map of interrelations among the graphical movement indicators and changes in the drawing elements in an increasing continuum of complexity and the roles conferred to figures sketched. The conclusion is that changes evaluated in drawings can interactively reflect mental representation, and they could promote its transformation. The applied transfer of the results to education is discussed, in order to optimize the representational complexity and cognitive development.

## 1. Introduction

Understanding human development and the successive emergence of competencies is exciting, because it allows us to grasp the complexity of the processes involved in the organic base (the brain), from which the human mind progressively emerges, along with the progressive knowledge of the world, the resolution of tasks, language and decision making, among other skills.

One of the most promising current interdisciplinary approaches to investigate and understand cognitive development is neuroconstructivism [1], a framework that unifies different disciplines such as neuroscience, computational models, Piagetian constructivism and epigenetics to describe how the brain is transformed and specializes throughout the development trajectory. Development is understood as a set of continuous changes resulting from the covariation between biological and environmental constraints, through which increasingly complex representations of reality are constructed, along an individual probabilistic epigenesis during which the human mind emerges, originating from the brain in transformation [2].

The internal representations of physical and social reality built from the beginning of life are progressively modulated and expressed through external representational systems such as drawing or writing, which allow us to manifest what is known about the environment, while being influenced by them. Therefore, representations about the world are dependent on the environment they model [2], while they are processed and stored through neural functioning.

One way to study mental representation is through the external manifestation of this internalized reality, and a child’s drawing is the first notation that allows it, showing how and what is known about reality. Despite the fact that movement is an inherent possibility of objects and situations that the newborn preferentially captures, allowing an object and its movements to be identified, as well as its shape and position [3,4,5,6], there are few studies on how the representation of the movement of people and objects is expressed through drawing [7], despite the fact that part of the contents of the first dynamic representations revolve around the identity of events, objects and people, as well as their movement and position [8]. These are expressed from the first scribbles drawn from the first year of life, which are enriched towards a controlled and meaningful form of drawing.

These initial graphic representation modes of forms (configurative) and actions (dynamic) interact with each other, expressing events as continuous processes in time and space. With their development, children represent their thinking about shapes and movements through a series of drawings, or in one, and sometimes with functional dependence between the description of the movement and the description of the form, a reflection that their representation is composed by coding various aspects of the object or event, represented in different axes or vectors. Other times, it is the result of knowledge about how to represent them [8].

The significant representation of movement in children’s drawing emerges between the ages of 5 and 6 [9,10], implying the appearance of various resources [11]: (a) partial deformations of a segment of the body; (b) redundancy in details; (c) modification of the habitual position of the body or object, or parts of them; (d) marking as horizontally drawn lines or arrows, and (e) representation in profile of the moving element. Its achievement indicates the end of a simple juxtaposition, becoming a technical resource that establishes a system of relationships between the elements of the drawing.

Among the existing qualitative studies, those carried out [12,13] stand out for their thoroughness in the profusion of movement indicators. Their results indicate that movement is one of the main causes of the change in graphic representation after 5 years. The need to express action encourages the modification of the method to make the drawing, altering the orientation, shape and global conception of it and involving new formulas and technical resources that give rise to progressive transformations; this is more significant after 6–7 years of age [13].

### Executive Functioning in Movement Drawing

Recently, the development of drawing has been related to executive functioning and its core components [14], highlighting the role of cognitive flexibility similar to the representative redescription (RR) [15]. In the same way, ref. [16] opened a line of research on children’s ability to modify drawings that has led to other studies on the development of flexibility through two types of tasks that show graphic transformation: (a) adding non-existent objects and (b) representing movement [17,18,19,20,21,22].

The results indicate that children from 4 to 6 years old are capable of drawing differences in the activity of the figures, adding new distinctive characteristics from ages 6 to 10, which could be facilitated by instructions from ages 7 to 9 [23,24]. With this, the proposals of [16,25] are corroborated: from the age of 5, the distinction between static and running figures begins, drawing these with longer and more separated legs, while at the age of 7, adaptations include profile orientation of the face and feet, curvature in the arms and legs, and lengthening of the legs [23].

On the other hand, ref. [7] found that, although from ages 4 to 6 a totally rigid drawing procedure is not used, beyond that age, postulating that rigidity does not predict the children’s ability to manipulate representations, nor is it associated with final interruptions. That is, although flexibility improves, stiffness continues to be present [26]. It is necessary to distinguish between procedural rigidity and rigidity from procedure as the natural order of the drawing. While the first is a cognitive constraint by which the child creates a facilitating routine graphically, the rigidity of the procedure would be a cognitive limitation, with the usual routines and the natural drawing order explaining that the elements of the children’s graphic representations are produced sequentially fixed [22]. Consequently, there is no rigid routine in children’s drawing, nor are the levels of rigidity related to age, being similar from ages 4 to 8 [27] and being linked to fundamental aspects of representation both at the spatial location of the elements as well as at the individual trace level.

The changes in drawing involve three resources: the availability of external models, the endogenous changes in mental representations and the theory of the graphic representation of inexperienced children [28]. This suggests that the working memory conditions the quantity of information subject to the RR process; that is, gradual changes in the information processing capacity determine how much information can be used. The development of other components of executive functions may also explain why the development pattern of younger children is slower and more gradual [29], while the improvement of executive functions facilitates the RR process.

Conceptual and procedural flexibility can emerge earlier than expected [26]. Throughout preschool development, the child’s ability to modify graphic “stereotypes” increases; that is, there is an improvement in the flexibility of the graphic representation of reality [21,25,27]. According to [30], the flexibility of the drawing depends on (a) the amount of attention resources (M capacity) that a child can use to activate operational and figurative schemes in the relevant tasks; (b) the automatic activation of figurative schemes of perceptual inputs, and (c) the activation of executive plans that set adequate objectives and control performance. Consequently, RR needs to adopt an integrative form that takes into account changes in information processing in terms of control, executive functioning, and task complexity [20,24]. Added to this is the role played by the biomechanical, cognitive and contextual constraints [31].

In summary, drawing is a complex skill that involves biomechanical, graphomotor, perceptual, cognitive and social developmental skills [31] and is the result of multiple factors, such as attention to detail [32], working memory [30], inhibition [21,33], flexibility [22] and the underlying representational processes [18,22,34].

Thus, the study of the emergence and development of movement in children’s drawing is an instrument that allows us to evaluate the graphic representational processes as indicators of the changes in the child’s cognitive development, a functional reflection of the transformations in their mental representation of reality, especially between 5 to 8 years of age. Therefore, capturing the signs of movement in a continuum of changes and transformations will allow inferring the underlying representational mental restructuring [4,35].

In accordance with the above, the general objective of this work is to evaluate the differences in the objective indicators used by children from 5 to 8 years of age to represent movement in drawing activities, through the analysis of polar coordinates that will allow knowledge of the continuum from its non-representation towards full representation, as well as evaluating the appearance and role of movement indicators that establish the step between the two through the vector representation of the associations between them, in each of the ages under study.

## 2. Materials and Methods

### 2.1. Sample

Participants were drawn by non-probabilistic intentional sampling among students from the second year of Early Childhood Education to the third year of Primary Education. The final sample was composed of N = 240 participants with typical development, distributed according to the age range of the study from 5 to 8 years, with n = 60 participants per age group. A total of 240 drawings were collected, with a total of 480 characters analyzed.

### 2.2. Instruments 

The stimulus material, adapted from [13] and extracted from [36], consisted of the narration of a story that explicitly suggests action scenes close to the children and claims to represent movement graphically. Two characters, *wolf and rabbit*, have their own movement, characterized by the escape of the rabbit, chased by the hungry wolf. (For an in-depth description, see Appendix A. Examples of drawings made by children in the three macrocategories of study: (a) static; (b) indication; (c) movement).

To capture the movement in the drawings, a coding instrument consisting of a mixed system of field formats and category systems was adapted ad hoc; all category systems met the requirement of completeness and mutual exclusivity [37,38], configured by the following components: (a) n = 3 macrocategories (*static, indication and movement*); (b) n = 12 categories that comprise different *positions-orientations* and (c) n = 49 microcategories, such as *body and external indicators*.

Regarding the quality control of the data, both Cohen’s kappa coefficient and the generalizability study revealed that the reliability was optimal, and a highly significant goodness of fit was estimated regarding the validity of the coding instrument. This confirmed the consistency of the instrument [36]. Due to its breadth, a simplified version was made with codes that presented greater significance in the analysis of polar coordinates (Table 1).

### 2.3. Design

The administration was carried out in each school group (2nd and 3rd of Early Childhood Education; 1st, 2nd and 3rd of Primary Education) during school hours. The administration interval ranged between 60 and 75 min. As the procedure, the story was told, focusing on the aspects of movement, and at the end the children were asked to draw it. For all participants, parents provided written informed consent in order for their child to participate in the study. Participants were informed of the confidentiality of their responses and of the voluntary nature of the study. No incentive was provided for their participation. The investigation was in accordance with the Declaration of Helsinki.

The proposed observational design was Nomothetic/Punctual/Multidimensional [37]. It was nomothetic due to the plurality of participating units, punctual because each participant made a single drawing of the wolf chasing the rabbit, and multidimensional because, from the theoretical framework, a proposal of dimensions and subdimensions was made that supported the observation instrument.

### 2.4. Data Analysis

The analysis of polar coordinates was proposed by Sackett [39]; it allows obtaining a map of interrelationships between the codes (behaviors/categories) of an observational record and also representing them graphically using vectors. The analytical technique of polar coordinates responds to a double strategy of data reduction and vector representation of the complex network of interrelations that are established between the different categories corresponding to each of the dimensions established in the observation instrument. Its application has increased in recent years in different fields, especially in those of developmental psychology, clinical psychology, social psychology and sports.

To carry out a polar coordinate analysis, a lag sequential analysis must be previously performed [40], and the adjusted residuals obtained constitute the data for the polar coordinate analysis. Based on the adjusted residue values obtained in the lag sequential analysis, the corresponding z scores are found, as relative indices of sequential dependence, both when the focal behavior acts as a criterion and when it acts as conditioned, and both prospectively (lags +1, +2, …) and retrospectively (lags −1, −2, …). It must be undertaken with at least 5 lags (from −5 to −1 and from +1 to +5), according to Sackett [39]. The basic structure of the polar coordinate technique complements prospective sequential analysis with positive lags and retrospective analysis with negative lags [40].

For each analysis of polar coordinates, the proposal of a focal behavior is required, which is assumed, according to the study objectives, as the generator or initializer of a series of connections with the other categories, called matching or conditioned behaviors. In this study, given the breadth of the observation instrument, multiple polar coordinate analyses were carried out, specifying the respective focal and conditioned behaviors.

The polar coordinate analysis complements the prospective (forward feeding) and retrospective (backward feeding) perspective, allowing us to know the relationships between focal behavior and conditioned behaviors. Therefore, the analysis is based on both perspectives. Sackett [41] ingeniously applied the Z_sum_ statistic proposed by Cochran [39], providing a powerful means of data reduction when the data are independent. According to [39], the sum of N independent z scores is normally distributed, with X = 0 and σ $=\sqrt{n}$ (where n is the number of lags), so that $Z_{\mathrm{sum}}=\sum_{1}^{n}\frac{z}{\sqrt{n}}$, which allows us to measure the associative strength or consistency between several behaviors.

We applied this to the obtained adjusted residual values (which are independent of each other because they each respond to a different calculation, given that the lags are different), considering the criterion behavior of the sequential behavior and the conditional behaviors in positive lags to obtain the prospective Z_sum_ values. In addition, we applied the same procedure (but using conditional behaviors in negative rather that positive lags) to obtain the retrospective Z_sum_ values. In both cases, we applied the binomial test between conditional probabilities (from the coding of the drawings) and expected or unconditional probabilities (random effect). 

From the Z_sum_ values obtained, the length and angle of each of the vectors are found, which allow graphing the interrelation between the focal behavior and each of the conditioned behaviors. The length of vectors is the distance between the origin (0,0) of coordinates Z_sum_ and the intersection point, which corresponds to $\sqrt{X^{2}+Y^{2}}$, where X is the Z_sum_ corresponding to the focal behavior and Y to the conditioned behavior, as well as the angle, whose trigonometric function of the sine arc is Arc sin =$\;\frac{Y}{length},\;$after taking into account the number of degrees to be added or subtracted in the different quadrants (see below), while indicating the type of relationships established.

The interpretation of these vectors is carried out taking into account their angle and, consequently, the quadrant in which they are located, which indicates the nature of the interrelation between the focal behavior and each conditioned behavior, and taking into account their length, which indicates the presence or absence of strength (statistical significance) of the association between focal and conditional behaviors.

Each prospective and retrospective Z_sum_ can have a positive or negative sign, so the “game” of signs when combining the prospective and retrospective values for each focal behavior will determine in which of the four possible quadrants (I, II, III, IV) the vectors corresponding to each of the conditioned behaviors will be located, in accordance with the following provision:

*Quadrant I* (+ +). The focal and conditional behaviors activate each other. Angle: φ = (0 < φ < 90). 

*Quadrant II* (− +). The focal behavior inhibits and is activated by the conditional behavior. Angle: 180 − φ = (90 < φ < 180).

*Quadrant III* (− −). The focal and conditional behaviors inhibit each other. Angle: 180 + φ = (180 < φ < 270). 

*Quadrant IV* (+ −). The focal behavior activates and is inhibited by the conditional behavior. Angle: 360 − φ = (270 < φ < 360). 

Ultimately, all the vectors corresponding to each of the conditioned behaviors in relation to the focal behavior considered can be drawn, as well as the selection of the significant or very significant ones, as has been done in this study. Thus, a powerful statistical technique is incorporated that allows to methodologically solve those of the observation instrument.

The analysis of polar coordinates is carried out using the free program HOISAN 2.0 [42] (University of Málaga, Malaga, Spain), and a graphic optimization of the vectors is achieved using the free program R [43] (https://jairodmed.shinyapps.io/HOISAN_to_R/ access on 30 June 2019).

## 3. Results

The results report stable patterns of behavior (or graphic indicators) in the representation of movement in the drawing between the selected focal behaviors and the rest of the behaviors collected in the observation instrument as macrocategories, categories and microcategories. Given the number of behaviors and the profusion of significant relationships, the focal behaviors with the most significant vectors were selected, which allows grasping the underlying representational processes through external graphic indicators of changes in children’s cognitive development. (For an in-depth description, see Appendix A. Indicators of graphic representation of movement with very significant relationships (*p* < 0.001) between focal and conditione behavior).

These results are presented based on (a) age, considering the highest number of significant relationships in each of the study categories; (b) who is the most active character in the drawn story.

At 5 years of age, the macrocategory *static* predominates with the rabbit and the *movement* macrocategory predominates with the wolf in the drawing. Within the *static* macrocategory, the relationships between the graphical indicators of movement that are mutually activated predominate; in the *indication* macrocategory, the focal activates and the conditioned one inhibits; in the *movement* macrocategory, the focal inhibits and the conditioned one activates.

Figure 1 shows that in the representation of the macrocategory *static*, the character with the highest number of significant vectors is the rabbit, with six significant relationships. In the behavioral map estimated from the focal behavior *vertical-front crossed arms* (*evfbcr*), it is observed that the relationship of prospective and retrospective mutual activation with three conditioned behaviors predominates (*evfcf*; *evfpj evfor*) (see Table 1 for codes). On the other hand, in the graphic representation of the *indication* macrocategory, the wolf is the character with the most significant relationships, specifically, the *total profile static indicators* focal behavior is activated (*ihptice*), and the conditioned one is inhibited (*ihpcficecg*).

In the representation according to the macrocategory *movement*, it is also the wolf that reports the most significant relationships between the graphical indicators. The most represented relationships are retrospective activation and prospective inhibition between the focal behavior *vertical-front face profile* (*mvfcp*) and the conditioned ones (*mvfbed, mvfps, mvffpc, mvptbed, mvptr, mhptr*).

At 6 years, the graphic representation of the predominant movement (plus significant vectors) is that of *static* and *indication* in the rabbit and the macrocategory of *movement* in the wolf. Relationships in which focal and conditioned behavior are mutually activated predominate in the three macrocategories of *static, indication and movement*, as reflected in Figure 2. As at five years, the greatest number of significant relationships occurs in the situation in which the focal behavior *vertical-front dropped arms* (*vfbc*) is mutually activated, with five conditioned behaviors (*vfbc, vfcf, vfbc, vfpj, vfpb, vfcc*). In the representation of the macrocategory *indication*, the rabbit is the character with the most reported indicators; the focal behavior with the most significant relationships is *static body indicators with scribbles* (*ivficecg*), with two conditioned behaviors: one relationship with mutual activation with (*ivficecg*) and the other with mutual inhibition with (*ivpcficecg*).

Regarding the representation through the macrocategory of *movement*, it is still the wolf that presents the most significant relationships for focal behavior *horizontal profile face front ears in the air* (*mhpcfoa*), whose greatest representation occurs between the relationships in which both behaviors are mutually activated prospectively and retrospectively (*mvfbed*, *mhpcfps*, *mhpcfped*, *mhpcfoa*, *mhptps*, *mhptped*, *mhptpea*).

At 7 years of age, the predominant representation in both characters is through the *movement* macrocategory, significantly and mutually activating the relationships between focal and conditioned behavior. On the other hand, when the graphic representation is made according to the macrocategories of *static* and *indication*, the focal and conditioned behaviors mutually inhibit each other. The character with the most significant relationships is the wolf (see Figure 3) in the macrocategories of *static* and *indication* and the rabbit in that of *movement*, indicating the greatest dynamism in the drawing of the story.

Figure 3 also shows that, at 7 years, the most significant focal behavior is *vertical-front face front* (*evfcf*), but the one with the most significant vectors is the relationship in which both mutually inhibit each other (*evfcf*, *evfpj*, *evfcc*). In the graphic representation using the *indication* macrocategory, the wolf offers more significance between the focal behavior *horizontal profile face front static body indicators with scribbles* (*ihpcficecg*) and the conditioned behavior (*ihpcficecg*), which mutually inhibit each other. On the other hand, in the graphic representation using the *movement* macrocategory, the greatest number of significant vectors appears in the rabbit drawing between the focal behavior *vertical arm stretched out in front* (*mvfbea*) and the conditioned behaviors, with mutual prospective and retrospective activation with five conditioned behaviors (*mvfbea*, *mhpcfped*, *mhpcffpc*, *mhpcfppe*, *mhptoa*).

Finally, at age 8, the predominant graphic representation is through the *movement* macrocategory. Statistically significant relationships are reported with a predominance in which the focal and conditioned behavior are mutually activated, which denotes that at this age the child has a greater number of resources to graphically express movement.

Figure 4 shows that, at this age, in the representation of the drawing using the macrocategories *static* and *indication*, there are no significant relationships between behaviors (or graphic indicators), indicating their marked decrease as a means of representation. The character with the greatest statistical significance in the graphic indicators within the *movement* macrocategory is the rabbit, who acquires the greatest prominence in the drawing of history, with the predominant focal behavior of *vertical front arm joint flexion* (*mvffba*) and with a high incidence of prospective and retrospective mutual activation, with five conditioned behaviors (*mvffba*, *mvpcfbed*, *mvpcfppe*, *mvptped*, *mvptfpa*).

Table 2 lists the macrocategories and categories in which movement is represented in the drawings, for each age of study and depending on each character.

As can be seen, at 5 years, the *static* macrocategory predominates in the rabbit, with behaviors that are mutually activated, while the *indication* and *movement* macrocategories have less presence; they appear only in the wolf and are activated and inhibited between them. At 6 years of age, the three macrocategories of study are mutually activated; the participants draw more in *static* and *indication*, but at the same time there is an increase in graphic representation through the macrocategory of *movement* compared to the previous age. On the other hand, at age 7, there is a predominance in the drawing of the *movement* macrocategory, followed by the *indication* one; a mutual inhibition between *static* and *indication* and a mutual activation in the *movement* macrocategory is verified. At 8 years of age, the graphic representation appears mostly through the *movement* macrocategory, not presenting significant relationships in those of *static* and *indication*; in the *movement* macrocategory, focal and conditioned behavior are mutually activated.

Consequently, the evaluation of these graphic indicators shows that there is a change in the role of the characters in the story, depending on the age of the participants.

As can be seen in Table 2, there are statistically significant differences in the representation of movement in the study ages from its representation with static indicators towards those of movement with change in the dynamism of the characters in the story, first with the greater activity of the wolf (chaser), and then with the greater activity of the rabbit (flight behavior). While at 5 and 6 years, the wolf is more active and causes more *movement* and *indication* in the rabbit, at 7 and 8 years it is the rabbit that takes an active role, eliciting more *movement* in the wolf.

These results show that it is possible to evaluate changes in the graphic representation of more flexible movement and according to the role of the characters in a story, through a sequence in the drawing that reflects transformations in the graphic representation of greater activity among the characters, according to the role of persecutor (initially more significant) or flight (at an older age). The production of signs in the drawing, with interaction of complex focal behaviors, provoke indicators of more elaborate graphic behaviors, which lead to increased flexibility in the representation of the movement with age, introducing new, more complex graphical indicators that also establish statistically significant relationships that trigger other behaviors.

## 4. Discussion

The results obtained offer resources to evaluate the presence and meaning of graphic indicators in children’s drawings through the way they draw movement, which can reflect changes in mental representation from 5 to 8 years old and, therefore, their derivations for learning.

On the one hand, the results obtained corroborate that the need to represent movement in the characters of a story leads children to develop new resources that progressively transform their graphic scheme [25,44]. Likewise, they confirm that the representation of movement in children’s drawings is possible at 5 years of age and follows a development process marked by a progressive increase in its complexity and realism up to 8 years; that is, children are capable of modifying their habitual representation schemes from an early age, corroborating the findings of studies such as [21,45].

Specifically, the graphic representation of the movement follows a continuum from the predominance of its representation with *static* indicators (that is, figures without movement) through to the increase of its representation with indicators of *indication*, until those of *movement* predominate, accompanied by a change to a greater mutual activation in the macrocategories in which the child has more strategies to draw; that is, at 5 years they predominate in *static*, at 6 in *indication* and at 7 and 8 in *movement*.

On the other hand, another interesting question derives from which are the indicators of the graphical representation of movement that promote relationships with other indicators at different ages because they allow evaluating the continuum of the transformations that take place and the introduction or the passage to other more complex indicators, flexibly; that is, what is the macrocategory that appears as the focal behavior that promotes more significant relationships with other conditioned behaviors (or graphic indicators), giving rise to more elaborate ones? According to the results, the progression goes from drawing the *face profile* at 5 years to *face front profile* at 6, while at 7 and 8 the focal behaviors that elicit the most movement are the modification of the limbs but with more elaborate strategies by older children; specifically, at 7 years it is *arm stretched out in front*, the one that causes more infrequent conditioned behaviors at younger ages, such as *curvature of the arm*, which at age 8 became *arm joint flexion* as the most complex behavior evaluated in the drawings. Consequently, the graphical indicators that cause a greater representation of movement follow a growing trend of development, becoming progressively richer. This allowed us to verify which are the behaviors of graphical representation of movement that promote and facilitate the increase of movement indicators, influencing the representational changes in each of the study ages, which facilitates a differential practical application according to the moment of development.

These results are consistent with previous studies that postulate that the arms and legs are the elements in which the greatest degree of modification is observed through more diverse solutions and that the inclination of the trunk or the flexion of the limbs of the figure to express movement are the most difficult indicators in their graphic representation [21,25,30]. In turn, these results provide a new proposal, which modifies that of [44,46] regarding the emergence at earlier ages of the presence in the drawing of the orientation of the profile of the face and the feet of the characters.

Another contribution of the evaluation is to know which categories of graphical indicators facilitate a better graphical representation of movement. It is interesting to note that at 5 years of age the representation predominates in the *indication* and *movement* macrocategories in the profile position of the characters, in contrast to the horizontal position for the *indication* macrocategory. On the other hand, the front orientation in the drawing is significantly related to the representation of the *static* and *movement* macrocategories, while in the *indication* macrocategory it is the representation of the *profile face front*; this is in accordance with the results of [7] regarding that transformations that require change in spatial location are more difficult for children to modify. Along the same lines, [21,28] postulate the difficulty of children to make complex inter- or intra-representational spontaneous changes in drawing, such as in the position or orientation of moving figures.

Finally, the results suggest that these transformations in the graphic representation of movement are conditioned by the content of the story, as the characteristics of the characters differentially motivate children to represent the characters with more or less movement, according to age.

Specifically, at 5 and 6 years of age, the youngest draw more movement for the wolf, the figure that chases, and at 7 and 8 years it is the rabbit that runs away who is represented with more movement, corroborating the postulates of other authors such as [20] regarding the characters, and those that suggest the influence of the proximity of content or of the instructions on the activity of the characters [23] as facilitators of change.

## 5. Conclusions

Among the conclusions, it stands out that drawing the movement of characters facilitates or is related to the representational change of reality, since the need to graphically express an action encourages the modification of the method that children use to draw movement [24,31]. This implies that when the external representation system is reconstructed, the internal cognitive representation is also reconstructed in an interaction between them and their mutual reconstruction [44,47].

Therefore, evaluating the changes in the representation of movement in children’s drawings is a key aspect in the study of the acquisition of new mental representations and has important practical consequences, but it is not enough to know the isolated data of a certain behavior [36]. For its practical application, it is important to go further and analyze the behaviors that act as antecedents and consequences to decipher the how and why of the changes in the representation of movement in children’s drawings, for which the polar coordinates technique is shown to be effective, using it to verify the existence of different patterns of the graphic representation of movement in children’s drawings and their relationships, comparatively between different ages.

The results point to statistically significant differences that support showing which are the transformations in children’s drawings as well as the adjustments in the graphic representation of movement that can indicate progress in the management of internal representations. This is thanks to the changes that reflect external representation procedures, in accordance with neuroconstructivist postulates on cognitive development, and the role of partial representations facilitating more complex changes determined by proactivity and progressive specialization [48].

In summary, this work opens a path for didactic application for the evaluation and facilitators of representational transformations in children’s drawings from 5–6 years of age that, functionally, can promote the redefinition of internal representation, that is, cognitive change on the knowledge of reality and its organization.

In conclusion, this study has made it possible to know the behaviors that act as antecedents and consequences to decipher the how and why of changes in the representation of movement in children’s drawings, allowing us to go beyond the isolated data of the differences that occur depending on the age of a certain behavior; this allows its practical application by training the behaviors that promote changes to influence the restructuring of mental representation.

## Figures and Tables

**Figure 1 ijerph-19-02844-f001:**
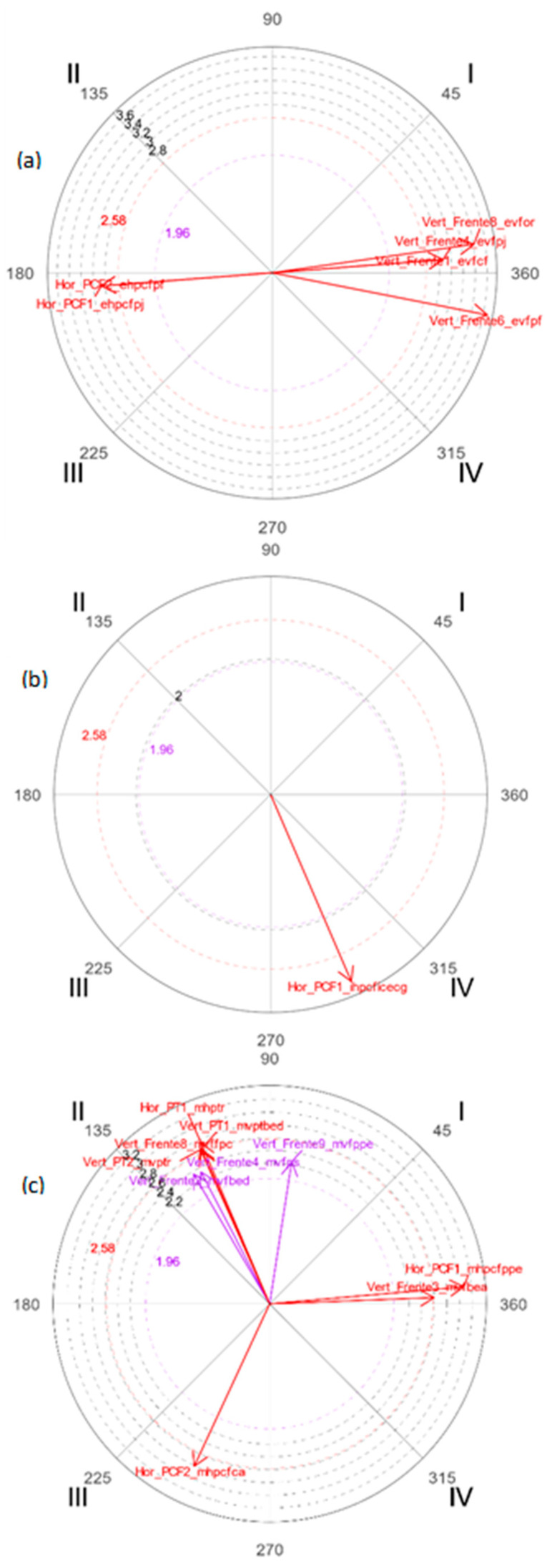
Polar coordinate maps of the character with the highest number of significant vectors in the representation of movement in the drawings at 5 years. (**a**) Macrocategory *static*-rabbit; (**b**) macrocategory *indication*-wolf; (**c**) macrocategory *movement*-wolf.

**Figure 2 ijerph-19-02844-f002:**
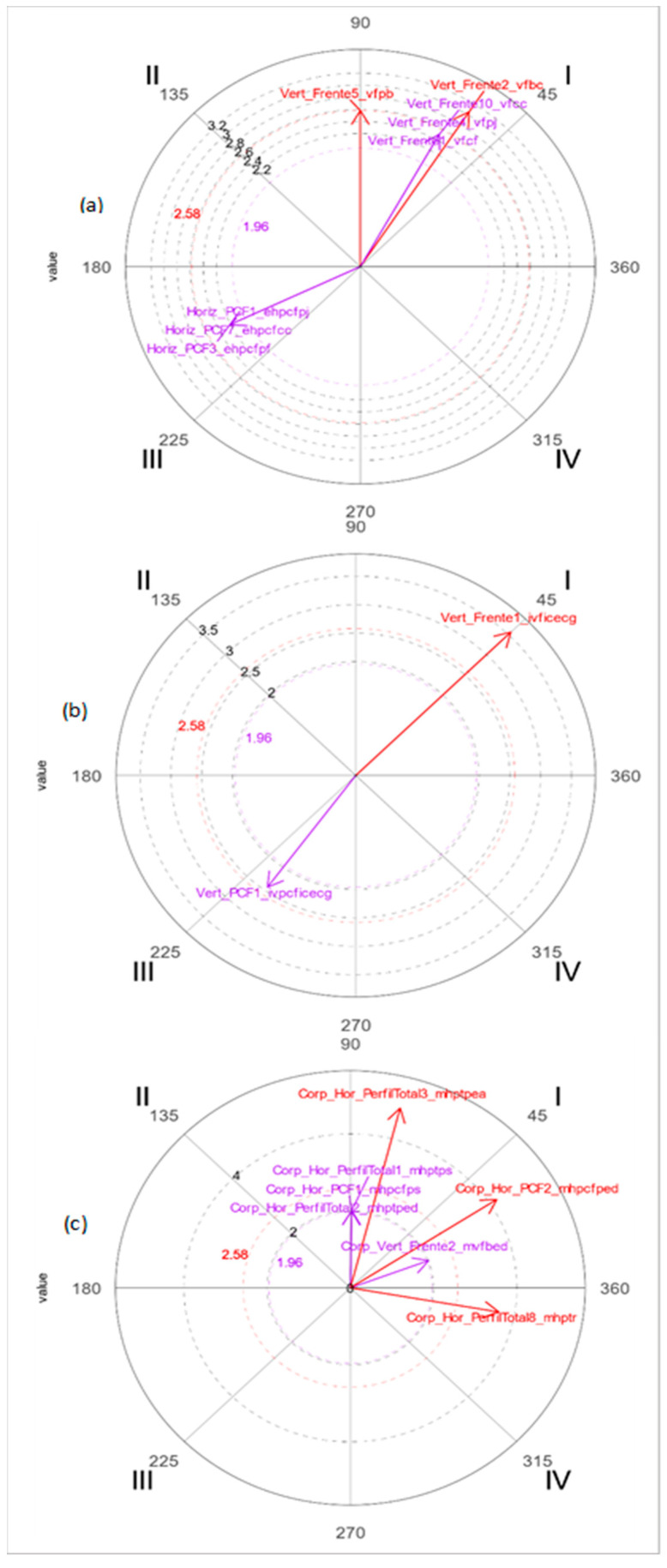
Polar coordinate maps of the character with the highest number of significant vectors in the representation of movement in the drawings at 6 years. (**a**) Macrocategory *static*-wolf; (**b**) macrocategory *indication*-rabbit; (**c**) macrocategory *movement*-wolf.

**Figure 3 ijerph-19-02844-f003:**
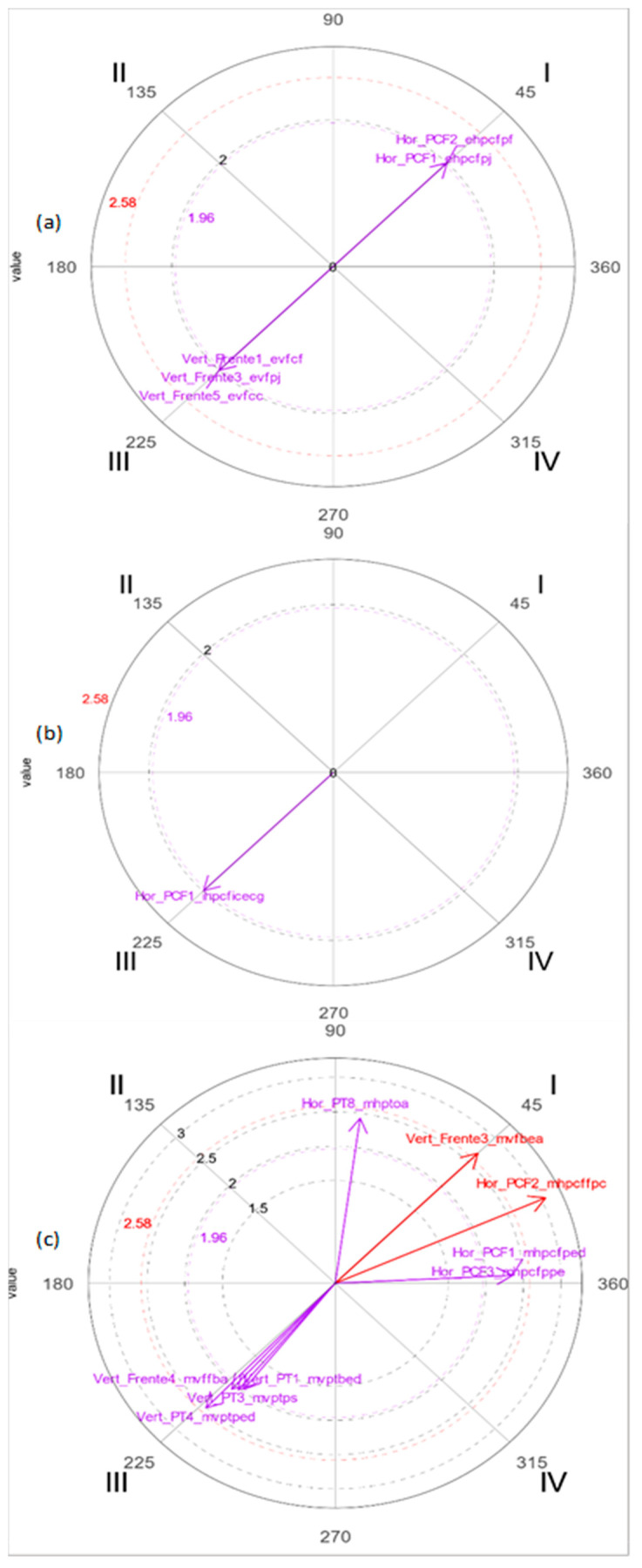
Polar coordinate maps of the character with the highest number of significant vectors in the representation of movement in the drawings at 7 years. (**a**) Macrocategory *static*-wolf; (**b**) macrocategory *indication*-rabbit; (**c**) macrocategory *movement*-rabbit.

**Figure 4 ijerph-19-02844-f004:**
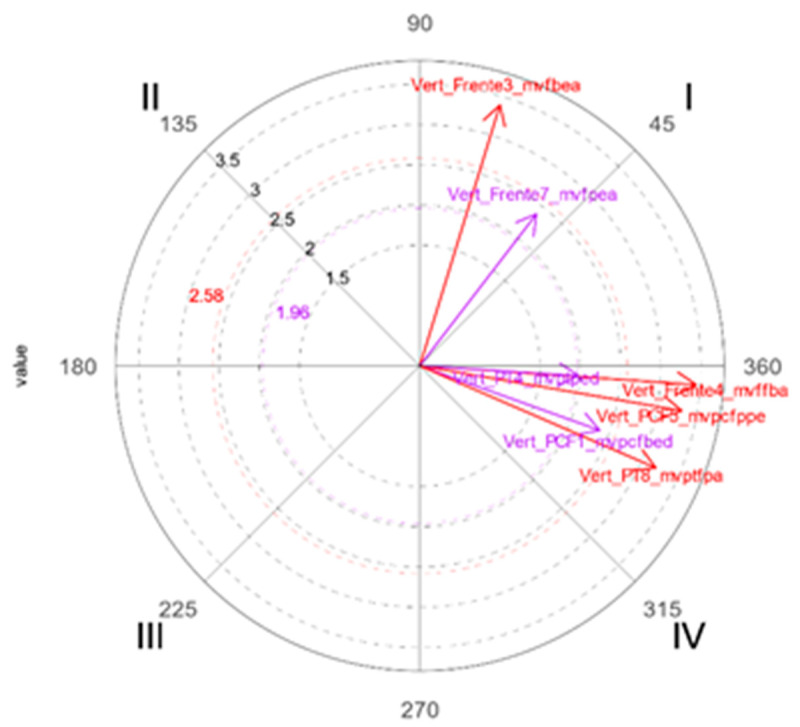
Polar coordinate map of the rabbit character with the highest number of significant vectors in the representation of the movement macrocategory in the drawings at 8 years of age.

**Table 1 ijerph-19-02844-t001:** Simplified version of the data collection instrument: codes with the highest number of significant vectors.

Macro Category	Category	Microcategory	Code
STATIC	Vertical-Front	Face front	Vfcf
		Dropped arms	Vfbc
		Crossed arms	vfbcr
		Legs (together)	Vfpj
		Feet front	Vfpf
		Straight ears	Vfor
		Fallen tail	Vfcc
	Horizontal Face Front Profile	Legs (together)	Ehpcfpj
		Feet front	ehpcfpf
INDICATION	Vertical-Front	Static body indicators with scribbles	ivficecg
	Vertical-Back	Static body indicators with scribbles	iveicecg
	Horizontal Face Front Profile	Static body indicators with scribbles	ihpcficecg
	Complete Horizontal Profile	Static body indicators	Ihptice
MOVEMENT	Vertical-Front	Face profile	Mvfcp
		Arm stretched out in front	Mvfbed
		Arm stretched back	Mvfbea
		Arm joint flexion	Mvffba
		Legs (separated)	Mvfps
		Leg stretched back	Mvfpea
		Curvature leg flexion	Mvffpc
		Profile feet escaping	Mvfppe
	Vertical And Face Front Profile	Arm stretched out in front	mvpcfbed
		Profile feet escaping	mvpcfppe
	Complete Vertical Profile	Arm stretched out in front	mvptbed
		Leg stretched out in front	mvptped
		Leg joint flexion	mvptfpa
		Leg stretched back	mvptpea
		Wheel	Mvptr
		Legs (separated)	Mvptps
	Horizontal And Face Front Profile	Legs (separated)	mhpcfps
		Leg stretched back	mhpcfped
		Curvature leg flexion	mhpcffpc
		Profile feet escaping	mhpcfppe
		Ears in the air	mhpcfoa
		Tail in the air	mhpcfca
	Complete Horizontal Profile	Legs (separated)	Mhptps
		Leg stretched out in front	mhptped
		Leg stretched back	mhptpea
		Wheel	Mhptr
		Ears in the air	Mhptoa

**Table 2 ijerph-19-02844-t002:** Most significant vectors of the macrocategories in each age for the two characters according to the quadrant.

		Age 5			Age 6			Age 7			Age 8		
		Sta	Ind	Mov	Sta	Ind	Mov	Sta	Ind	Mov	Sta	Ind	Mov
Wolf	FA-CA	---	---	3	5	1	7	2	---	5	*---*	*---*	*---*
	FI-CA	---	---	6	---	----	---	---	---	---	---	---	---
	FI-CI	---	---	1	3	1	---	3	1	4	---	---	1
	FA-CI	---	1	---	---	----	1	---	---	---	---	---	6
Rabbit	FA-CA	3	---	3	---	---	---	---	---	2	*---*	*---*	2
	FI-CA	----	---	2	---	1	---	---	---		---	---	---
	FI-CI	2	---	3	3	1	---	---	---	3	---	---	---
	FA-CA	1	---	---	---	---	---	---	---	2	---	---	5

*Note.* FA = Focal activates; FI = Focal inhibits; CA = Conditioned activates; CI = Conditioned inhibits; Sta = static; Ind = indication; Mov = Movement. The numbers indicate the vectors with statistical significance (*p* = ≤ 0.001; *p* ≤ 0.005).

## Data Availability

Data are contained within the article.

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
