# Peer review of "Evaluation Using Polar Coordinate of the Representation of Movement in the Drawings of Children Aged 5 to 8 Years"

_ijerph, 2022, doi:10.3390/ijerph19052844_

Round 1

Reviewer 1 Report

 Evaluation using polar coordinates of the representation of movement in the drawing of children aged 5 to 8 years

The manuscript focuses on neuro-constructivism. In 240 children between the age of 5-8 years, the authors study the indicator used by children while drawing moving figures to understand the mental representation complexity from non-manifestation to fully expressed form.

Comments on abstract:

  1. The abstract is confusing and wordy. Simplifying the abstract to fewer words, simple sentences might help readers to understand the context better.
  2. The conclusion needs to be re-written in clear statements. At this point, the study outcome sounds very vague.

The study itself is very interesting, but the writing needs more attention.

  1. Line 9: “ Neuroconstructivismo” : Is this a word the authors are newly defining?

Comments:

The introduction can be more concise and to the point.

Line 48: do you mean " one way to understand mental representation of reality is through the external presentation of the internal manifestation of it, and child's drawing is the first notation that lets us understand their perception of the reality."

Line 51: I don’t think that movement is an inherent property of every object. So as for the situations, the authors need to define what is a movement in a situation.

Line 55: events and objects don't move, at least not voluntarily.

Line 58: do you mean figurative? Not sure what configurative means?

Line 75: The citations should be separated by “,” unless otherwise specified by the journal.

Line 75: incomplete sentence. stand out for their thoroughness about what?

Line 77: after 5 years of what? Be specific. [after the age of 5 years or after 5 years of age.]

Line 84: it would be extremely helpful if the authors mention someone’s work as last name et. al. opened a line of research ... [citation index].

Line 110: does the theory of the graphic representation of inexperienced children exclude the other resources mentioned?

Methods are not clearly defined.

Line 147:  while the method was appropriate, what was the criteria/ assumption for such a sampling strategy?

Line 163: what is the scale of measurements for each of these parameters. is it binary or a gradient scale?

Line 207: while referencing other papers for methods is a good idea, and often practiced, it is better to write the methodological details in simple words for the convenience of the readers.

Line 210: Define these terms in the current study context.

Line 228: what is mating behavior? In the current study context.

Line 234: What kind of power statistical technique was used? and to do what?

Line 238: Did the authors use any IDE or just the interface provided by the r-project?

In the result section, the Figures are of very poor quality and very difficult to understand.

  • Graphical presentation of some of the drawings as examples would have been helpful for a better understanding of the context.
  • It appears there are no strong conclusions from the study.
  • There might be better ways than polar plots to show vector representation (time-dependent directional representation with amplitude as the measure of maturity).
  • Line 390 – 400: very nice analysis of the results, but the authors might want to explain the context of maturity in children in generalized terms. It helps readers to understand the broader scope.

In 240 drawings were evaluated where each drawing had two characters: a wolf chasing a rabbit. It was not clear if the same kids were tracked over time, or these are independent kinds and just different groups as per the age. If this is just a different group of kids, there might be several parameters that influence the advancement in quality of drawing and attention to details as in each drawing. In fact, as we grow older, we generally do better at certain things. What is the author's interpretation of such change in behavior of different groups? Just saying polar plots are a better way to show the difference is probably not the best way.

Reviewer 2 Report

  1. Have you obtained approval from the ethics committee to give informed consent? If so, you should include the approval number.
  2. Were only parents informed when consent was obtained? Was no explanation given to the children?
  3. The results are probably overall, regardless of gender, but are there any effects due to gender differences, for example?

Round 2

Reviewer 1 Report

Most comments are appropriately addressed except the figure quality. I could not read the details in any of the polar plots. Please redo them as appropriate.

It very extremely inconvenient for readers to read illegible text.

Author Response

Thanks for the indication, we have changed the figures.